# Trusting in times of the COVID-19 crisis: Workplace and government trust and depressive symptoms among healthcare workers

## Research Article

**Cite this article:** Basic D, Czepiel D, Hoek HW, Martínez AM, McCormack C, Susser ES, Mascayano F, Moro MF, Carta MG, Martínez-Alés G, Fernández-Jiménez E, Barathie JAI, Karam EG, Nishi D, Asaoka H, Ayinde O, Gureje O, Afolabi O, Olaopa O, Ramírez J, Basagoitia A, S. Soto MT, Durand-Arias S, Šeblová J, Seblova D, Tenorio A, Ballester D, Burrone MS, Alvarado R, Santaella-Tenorio J, Ouali U, Isahakyan A, Lindert J, Sapag JC, Ramírez DE, Alnasser L, Rivera-Segarra E, Balalian A, Mediavilla R and Ven E (2025). Trusting in times of the COVID-19 crisis: Workplace and government trust and depressive symptoms among healthcare workers. *Cambridge Prisms: Global Mental Health*, **12**, e130, 1–11

**Keywords:**
healthcare workers; COVID-19; trust; depression; cross-country

**Corresponding author:**
Els van der Ven;
Email: e.m.a.vander.ven@vu.nl

Djordje Basic[1] , Diana Czepiel[1,2] , Hans W. Hoek[2,3,4], Adriana M. Martínez[4], Clare McCormack[4], Ezra S. Susser[4,5] , Franco Mascayano[4,5] , Maria F. Moro[4,6], Mauro G. Carta[6], Gonzalo Martínez-Alés[7,8,9,10], Eduardo Fernández-Jiménez[7,11], Josleen A.I Barathie[12], Elie G. Karam[12,13], Daisuke Nishi[14], Hiroki Asaoka[14], Olatunde Ayinde[15], Oye Gureje[15], Oyeyemi Afolabi[15], Olusegun Olaopa[15,16], Jorge Ramírez[17], Armando Basagoitia[18], María T. S. Soto[19], Sol Durand-Arias[20], Jana Šeblová[21], Dominika Seblova[21], Andrea Tenorio[22], Dinarte Ballester[23], María S. Burrone[24] , Rubén Alvarado[25], Julian Santaella-Tenorio[26], Uta Ouali[27,28], Anna Isahakyan[29], Jutta Lindert[30], Jaime C. Sapag[31], Dorian E. Ramírez[32], Lubna Alnasser[33], Eliut Rivera-Segarra[34] , Arin Balalian[35], Roberto Mediavilla[36] and Els van der Ven[1] 

[1]Department of Clinical, Neuro- and Developmental Psychology, Vrije Universiteit Amsterdam, Amsterdam, The Netherlands; [2]Parnassia Psychiatric Institute, Parnassia Groep, The Hague, The Netherlands; [3]Department of Psychiatry, University of Groningen, University Medical Center Groningen, Groningen, The Netherlands; [4]Mailman School of Public Health, Department of Epidemiology, Columbia University, New York, NY, USA; [5]New York State Psychiatric Institute, New York, NY, USA; [6]Department of Medical Sciences and Public Health, University of Cagliari, Cagliari, Italy; [7]Hospital La Paz, Institute for Health Research (IdiPAZ), Madrid, Spain; [8]Icahn School of Medicine at Mount Sinai, New York, NY, USA; [9]Department of Epidemiology, Harvard T.H. Chan School of Public Health, Boston, MA, USA; [10]Centre for Biomedical Research Network on Mental Health (CIBERSAM), Madrid, Spain; [11]Faculty of Law, Education and Humanities, Universidad Europea de Madrid, Villaviciosa de Odón, Spain; [12]Institute for Development, Research, Advocacy, and Applied Care (IDRAAC), Beirut, Lebanon; [13]Department of Psychiatry and Clinical Psychology, Saint George University of Beirut, Beirut, Lebanon; [14]Department of Mental Health, Graduate School of Medicine, The University of Tokyo, Tokyo, Japan; [15]Department of Psychiatry, University of Ibadan, Ibadan, Nigeria; [16]Department of Oral and Maxillofacial Surgery, University College Hospital, Ibadan, Nigeria; [17]School of Public Health, University of Chile, Santiago, Chile; [18]Chuquisaca Department, Consultora Salud Global, Chuquisaca, Bolivia; [19]Chuquisaca Department, Universidad San Francisco Xavier de Chuquisaca, Dirección de Investigación Ciencia y Tecnología, Sucre, Bolivia; [20]National Institute of Psychiatry Ramón de la Fuente Muñiz, Mexico City, Mexico; [21]Second Faculty of Medicine, Charles University Prague, Prague, Czech Republic; [22]Faculty of Medicine, São Paulo State University, Botucatu, Brazil; [23]University Hospital, Federal University of Rio Grande, Rio Grande, Brazil; [24]Institute of Health Sciences, Universidad de O'Higgins, Rancagua, Chile; [25]Interdisciplinary Center for Health Studies (CIESAL), Department of Public Health, School of Medicine, Faculty of Medicine, Universidad de Valparaiso, Valparaiso, Chile; [26]Department of Clinical Epidemiology and Biostatistics, Pontifical Xavierian University, Bogotá, Colombia; [27]Department Psychiatry A, Razi Hospital La Manouba, La Manouba, Tunisia; [28]Medical School of Tunis, University of Tunis El Manar, Tunis, Tunisia; [29]National Institute of Health Named After Academician S. Avdalbekyan, Yerevan, Armenia; [30]Faculty of Health and Social Work, University of Applied Sciences Emden/Leer, Emden, Germany; [31]Department of Public Health and Family Medicine, Pontificia Universidad Católica de Chile, Santiago, Chile; [32]Faculty of Medical Sciences, University of San Carlos of Guatemala, Guatemala City, Guatemala; [33]King Abdullah International Medical Research Center, King Saud Bin Abdulaziz University for Health Sciences, Riyadh, Saudi Arabia; [34]School of Behavioral and Brain Sciences, Ponce Research Institute, Ponce Health Sciences University, Ponce, Puerto Rico; [35]Question Driven Design and Analysis Group, New York, NY, USA and [36]Department of Psychiatry, Universidad Autónoma de Madrid, Madrid, Spain

## Abstract

Previous research has highlighted the negative impact of the COVID-19 pandemic on healthcare workers' (HCWs) mental health, yet protective factors remain underexplored. Emerging studies emphasize the importance of trust in government and interpersonal relationships in reducing infections and fostering positive vaccine attitudes. This study investigates the relationship between HCWs' trust in the workplace and government and depressive symptoms during the pandemic. The COVID-19 HEalth caRe wOrkErS study surveyed 32,410 HCWs from 22 countries, including clinical and nonclinical staff. Participants completed the Patient Health Questionnaire-9 and ad-hoc questions assessing trust in the workplace and government. Logistic regression and multilevel models examined associations between trust levels and depressive symptoms. High workplace trust (OR = 0.72 [0.68, 0.76]) and government trust (OR = 0.72 [0.69, 0.76]) were linked to lower odds of depressive symptoms, with significant between-country

variation. Country-level analyses showed that workplace trust was more protective in more developed countries and under stricter COVID-19 restrictions. Despite cross-country variation, HCWs with higher trust in the workplace and government had ~28% lower odds of experiencing depressive symptoms compared to those with lower trust. Promoting trust may help mitigate the mental health impact of future crises on HCWs.

## Impact statement

The coronavirus disease 2019 (COVID-19) pandemic placed extraordinary psychological and professional demands on healthcare workers (HCWs), leading to a widespread mental health crisis among those on the front lines. While numerous studies have examined risk factors for HCW distress, there remains a limited understanding of protective factors that may buffer against poor mental health – particularly at the institutional and policy levels. This study addresses that gap by demonstrating that healthcare workers who trust their workplace and government are significantly less likely to experience depressive symptoms.

Drawing on data from over 32,000 HCWs across 22 countries, our study offers the most comprehensive international evidence to date on the relationship between trust and mental health during a global health emergency. By identifying trust in government and the workplace as a cross-national protective factor, this research contributes to a deeper understanding of how psychological resilience can be fostered not only through individual coping strategies but also through systemic and organizational conditions that foster transparency, communication and perceived safety.

These findings have wide-reaching implications for governments, public health leaders and healthcare administrators. Promoting trust – through consistent communication, frontline worker inclusion in decision-making and visible institutional support – should be prioritized in emergency preparedness planning and workplace mental health strategies. Interventions that build and sustain trust may not only improve workforce morale but also help mitigate the psychological toll of future crises.

The impact of this work is both practical and global: it provides clear evidence that trust is not a vague ideal, but a measurable and meaningful component of health system resilience with direct implications for HCW mental health and the integrity of care delivery in times of crisis.

Since the coronavirus disease 2019 (COVID-19) pandemic was declared by the World Health Organization on March 11, 2020 (World Health Organization, 2020), policymakers have sought to enhance preparedness for future health crises (Kandel et al., 2020; Nuzzo et al., 2020). The pandemic placed HCWs under immense pressure, exposing them to elevated physical and psychological burdens, including viral exposure, long working hours, insufficient protective equipment and inadequate training (Li et al., 2021; Marvaldi et al., 2021; Phiri et al., 2021). Consequently, HCWs have experienced heightened rates of mental health problems, with depression prevalence reaching 43% in early pandemic months across 12 countries (Tong et al., 2023). Longitudinal evidence further suggests that symptoms often persist or worsen (Mediavilla et al., 2022), with consistent patterns observed across multi-country HEalth caRe wOrkErS (HEROES) data Cermakova et al., 2023; Asaoka et al., 2024; Moro et al., 2022).

Several risk factors for adverse mental health outcomes among HCWs have been identified, including female sex, younger age, frontline status, limited experience and repeated redeployment (Khajuria et al., 2021; Sun et al., 2021; Czepiel et al., 2024). Protective factors, such as professional experience, access to protective equipment and social support, have also been noted (Kisely et al., 2020). Despite a growing number of programs aimed at supporting HCWs, inconsistent implementation and limited clinical evaluation of these interventions – particularly among frontline workers – have left a gap in identifying effective, scalable strategies (Buselli et al., 2021; Delassalle and Cavaciuti, 2023).

Recent studies have drawn attention to the potential of institutional trust to mitigate pandemic impacts. Bollyky et al. (2022) demonstrated that trust in government and interpersonal relationships reduced infection rates across 177 countries. Among HCWs, institutional trust has been associated with improved mental well-being and confidence in public health protocols (El Sharif et al., 2022). Trust is also known to enhance healthcare quality and motivation (Okello and Gilson, 2015). During the H1N1 pandemic in Japan, HCWs reported that trust in institutions served as a work motivator, while those with low trust were more anxious (Imai, 2020).

While trust has been linked to HCW motivation and public health compliance, its role in shaping HCWs' mental health – especially during prolonged crises – remains underexplored. Most existing studies are limited to national contexts, constraining our understanding of cross-country disparities. Multinational studies are essential for identifying structural and contextual moderators of HCW mental health. Country-level indicators, such as the Human Development Index (HDI) and the Oxford Stringency Index (SI), provide tools to assess socioeconomic development and the strictness of pandemic-related government responses, respectively. Findings from the general population underscore the complexity of trust and stringency relationships. O'Hara et al. (2020) found that greater governmental stringency was associated with higher depression scores among those with strong trust. Conversely, Lee et al. (2021) observed that prompt, strict stringency measures correlated with lower depression prevalence.

In this study, we aimed to investigate the relationship between HCWs' trust in their workplace and government and depressive symptoms during the COVID-19 pandemic across countries of varying income levels. We also explored whether HDI and SI moderated these associations. We hypothesized that greater trust would be associated with fewer depressive symptoms, and that country-level factors would condition these relationships. By adopting a multilevel perspective, this study aims to deepen our understanding of how institutional trust and structural conditions interact to shape HCWs' mental health in global crises.

## Methods

### Study design and participants

This study is part of the broader COVID-19 HEROES project, a multinational initiative examining the impact of the COVID-19 pandemic on HCWs across 22 countries (Supplementary Figure S1), including both high-income and low- and middle-income countries (Mascayano et al., 2022). We used cross-sectional data from the first wave of the HEROES survey, collected between March 2020 and February 2021. This cross-sectional design was employed to capture a snapshot of HCWs' experiences during the early and most acute phases of the pandemic, a period marked by resource scarcity, rapidly evolving policies and heightened psychological distress, when institutional trust may have played a particularly critical role in shaping mental health outcomes.

Healthcare facilities were either randomly selected or pre-selected by national research teams, allowing the study to launch quickly across diverse healthcare systems. Recruitment strategies varied across countries and generally included three main approaches: (1) registers and databases (e.g., in Venezuela, potential participants came from three national registries managed by the Ministry of Health), (2) health facilities (e.g., in Belgium, 13 randomly selected hospitals) and (3) other institutions and organizations (e.g., in Guatemala, HCWs associated with academic institutions, union organizations and associations across the country). Nonprobabilistic or snowball sampling methods were used at most sites. Eligibility criteria included being: (1) ≥18 years of age, (2) employed in one of the preselected healthcare facilities that provide care to confirmed or suspected cases of COVID-19 and (3) having an internet connection. Both clinical (e.g., nurses and physicians) and nonclinical (e.g., cleaners and administration) HCWs were included. More details on each country's sample characteristics and recruitment strategies can be found elsewhere (Mascayano et al., 2022).

This study received approval from accredited Institutional Review Boards (IRBs) in each participating country, as well as the IRBs of Columbia University and the Faculty of Medicine, Universidad de Chile. To securely manage and protect the data, we used the digital platform provided by the Universidad de Chile, which is comparable to REDCap software (Harris et al., 2009, 2019). More details about data collection and security have been described elsewhere (Mascayano et al., 2022).

### Measures

#### Depressive symptoms

We measured depressive symptoms using the nine-item self-administered Patient Health Questionnaire (PHQ-9; Kroenke et al., 2001). This instrument assesses how often participants have experienced symptoms associated with depression in the last 2 weeks (e.g., "Little interest or pleasure in doing things"). Participants rated their answers using a 4-point Likert scale (0 = "*not at all*" to 3 = "*nearly every day*"), with a score ranging from 0 to 27. A cutoff score of 10 or higher indicated at least moderate depressive symptoms. This cutoff value has acceptable sensitivity (0.85) and specificity (0.89; Manea et al., 2012). The appropriateness of the standard cutoff score of 10 across diverse cultural contexts is further supported by a large individual participant data meta-analysis (Levis et al., 2019), which found no statistically significant differences in PHQ-9 diagnostic accuracy at this threshold across countries with varying HDI levels.

For secondary analyses, we used the sum of scores on all nine items as a continuous measure of depressive symptoms. The instrument has good internal reliability (Cronbach's $\alpha$ = .89; Kroenke et al., 2001).

It has been validated in different languages and populations (Gilbody et al., 2007; Sun et al., 2020). In the present study, the internal consistency reliability of the PHQ-9 was good (Cronbach's $\alpha$ = 0.89).

#### Trust factors

We measured trust in the workplace using a single-item ad-hoc question ("To what extent do you trust that your workplace can manage the COVID-19 pandemic?"). Participants used a 5-point Likert scale to rate their answers (0 = "*not at all*" to 4 = "*extremely*"). We measured trust in the government using a single-item ad-hoc question ("To what extent do you trust that your government can manage the COVID-19 pandemic?"). Participants rated their answers using the same 5-point Likert scale. For the primary analyses, we dichotomized trust variables into low trust (0–2: "*not at all*," "*slightly*" and "*moderately*") and high trust (3–4: "*considerably*" and "*extremely*"). This categorization aligned with prior HEROES reports using similar cut points to reflect meaningful distinctions in perceived institutional competence. Nonetheless, we recognize that dichotomization can lead to information loss and bias (Irwin and McClelland, 2003). Therefore, to assess robustness, we conducted sensitivity analysis using the original 5-point Likert scale. Results from these models are included in Supplementary Table S8 and showed substantively consistent patterns, supporting the robustness of the trust–depression association regardless of cut-point selection.

#### Moderators and covariates

We used ad-hoc questions to assess the following individual-level covariates: age (continuous), gender (0 = "*woman*," 1 = "*man*" and 2 = "*other gender*"), educational level, number of household members, current job and contact with COVID-19 patients ("During the past week, have you been close to patients who were suspected or confirmed cases of COVID-19?"). These have been identified as important sources of variation in mental health outcomes in other HEROES studies (Moro et al., 2022; Cermakova et al., 2023; Asaoka et al., 2024; Czepiel et al., 2024). Other potential covariates, such as occupational income level or prior mental health diagnoses, were not included due to a lack of measurement or high rates of missing data.

We included two country-level moderators: the HDI and the SI. The HDI, sourced from the 2020 UNDP Human Development Report (UNDP, 2020), reflects overall socioeconomic development by combining economic (gross national income per capita), educational and health indicators. Selected as a broad measure of structural and institutional capacity, the HDI is widely used in cross-national mental health research and captures dimensions theoretically relevant to population-level trust. The SI, based on the Oxford COVID-19 Government Response Tracker (Hale et al., 2021), is a composite measure of nine response indicators (e.g., school closures, workplace closures and travel bans), rescaled from 0 to 100 to indicate policy strictness. For each country, we calculated the average daily SI during its respective survey period to account for variation in government measures during data collection.

To account for pandemic severity across countries, we included infection-per-capita (IPC) and infection-fatality rate (IFR) estimates from Bollyky et al. (2022). These variables were standardized for relevant demographic, health and economic indicators. Detailed definitions and components of these indices are provided in the Supplementary Materials.

### Statistical analyses

All analyses were conducted using RStudio (Version 2022.07.2), with a significance threshold set at $p$ < .05. Descriptive statistics

were used to summarize key variables, including depressive symptom prevalence across levels of workplace and government trust. We used $\chi^2$-tests and *t*-tests to compare completers (no missing data on PHQ-9) and noncompleters on sociodemographic and trust variables.

Missing data were imputed under the missing at random (MAR) assumption, supported by results from Little's missing completely at random (MCAR) test, which rejected the hypothesis of MCAR (Hawkins $p < .001$; nonparametric $p < .001$). We used random forest-based multiple imputation via the MissRanger package. Following Oshiro et al. (2012), we specified 128 trees and achieved convergence within three iterations.

First, to examine associations between trust and depressive symptoms, we fit separate logistic regression models for workplace and government trust, with trust dichotomized into "low" (not at all, slightly and moderately) versus "high" (considerably and extremely). Models were adjusted for age, gender, educational level, number of household members, current job and exposure to COVID-19 patients.

Second, to evaluate baseline heterogeneity across countries, we first ran empty (null) multilevel models with random intercepts only. Intraclass correlation coefficients (ICCs) showed that 6.3% of the variance in depressive symptoms (PHQ-9) was attributable to between-country differences, justifying the inclusion of random intercepts by country in all subsequent models. Then, to account for country-level clustering, we conducted multilevel linear regression models using maximum likelihood estimation with country (22 countries; $n = 32,480$) as a random intercept (see Supplementary Table S3). These models included the same individual-level covariates, as well as country-level predictors: HDI, SI, IPC and IFR. As a robustness check, we also examined random slope models, allowing the effects of workplace and government trust to vary across countries. Model fit was compared using likelihood-ratio tests and Akaike Information Criteria (AIC).

Finally, we conducted moderation analyses to examine whether the associations between trust and depressive symptoms were moderated by HDI and SI. These models included interaction terms (e.g., trust × HDI) and were adjusted for all individual- and country-level covariates. We used the spotlight method to probe significant interactions.

## Results

### Sample characteristics

After excluding cases with missing PHQ-9 data, the final analytical sample included 24,782 participants. Completers were, on average, nearly 2 years older and lived with slightly fewer household members than noncompleters. They also reported higher trust in both workplace and government, with a significant difference in government trust (see Supplementary Table S1).

Participants had a mean age of 39.8 years (standard deviation [SD] = 11.2), and 72% identified as women. Most held undergraduate or postgraduate degrees (75%) and worked in hospitals or health centers (81%). Physicians comprised 27% of the sample, and nurses 20%. The prevalence of moderate or higher depressive symptoms was highest in Chile (31%) and lowest in Puerto Rico (3%). German HCWs showed the highest prevalence of workplace trust, with 58% indicating high trust, while Tunisian HCWs had the lowest at 13%. Similarly, Saudi Arabian HCWs reported the highest trust in government (60%), while Tunisian HCWs reported the lowest (3%). Among HCWs with at least moderate depressive symptoms, 63% ($N = 3,411$) reported high trust in their workplace, while 37%

($N = 2,009$) reported low trust. Conversely, regarding trust in the government, 37% ($N = 1,980$) reported high trust, whereas 63% ($N = 3,440$) reported low trust (Table A1). HDI estimates ranged from very high in Germany (HDI = 0.96) to low in Nigeria (HDI = 0.55), and average SI estimates ranged from 70.61 in Chile to 36.35 in Belgium (Supplementary Table S2).

### Association of trust with depressive symptoms

High workplace trust was associated with 28% lower odds of depressive symptoms (odds ratio [OR] = 0.72, 95% confidence interval [CI]: [0.68, 0.76]). Country-specific effects varied from OR = 0.84 (95% CI: [0.73, 0.96]) in Peru to OR = 0.31 (95% CI: [0.20, 0.46]) in Lebanon. Similarly, high government trust was associated with 28% lower odds of depressive symptoms (OR = 0.72, 95% CI: [0.69, 0.76]), with country-specific ORs ranging from 0.75 (95% CI: [0.64, 0.86]) in Peru to 0.15 (95% CI: [0.07, 0.33]) in Germany. Full country-specific estimates are reported in Table A2.

### Country-level effects and multilevel models

The empty models indicated modest clustering by country, with ICCs of 0.063 for PHQ-9, suggesting 6.3% variation in depressive symptoms across countries. Multilevel linear regression confirmed that trust in both the workplace and government remained robustly associated with lower depressive symptoms after accounting for individual- and country-level covariates, as well as country-level clustering (see Supplementary Table S3). HCWs reporting "extreme" workplace trust scored on average 2.34 points lower on the PHQ-9 than those with no trust (95% CI: [−2.64, −2.05]); the corresponding reduction for extreme government trust was 1.54 points (95% CI: [−1.94, −1.14]). Among the country-level predictors, none of the indices, including HDI, SI, IPC and IFR were significantly associated with depressive symptoms in the adjusted model. The variance attributed to country-level clustering remained modest ($\sigma^2 = 1.64$), with most variation occurring at the individual level ($\sigma^2 = 25.51$).

To test whether the association between trust and depressive symptoms varied across countries, we fitted additional random slope models in which the effects of trust were allowed to vary by country. Compared to the random intercept-only model, model fit significantly improved when allowing the slope of trust in the workplace to vary ($\Delta$AIC = 88, $\chi^2(2) = 91.8$, $p < .001$) and when allowing the slope of trust in government to vary ($\Delta$AIC = 10, $\chi^2(2) = 13.5$, $p = .001$). These findings indicate that the protective associations of trust with depressive symptoms are not consistent across countries, but instead vary depending on national and institutional context. Full model comparisons are presented in Supplementary Table S5.

### Moderation by country-level indicators

The interaction between workplace trust and HDI was significant, with the negative association with depressive symptoms stronger in more developed countries. Among participants from very high-HDI countries, higher workplace trust was associated with a steeper reduction in depressive symptoms ($B = -0.96$, 95% CI: [−1.35, −0.57], $p < .001$), compared to high-HDI ($B = -0.67$, 95% CI: [−1.06, −0.29], $p = .001$) and medium-HDI countries ($B = -0.67$, 95% CI: [−1.08, −0.26], $p = .001$ The association was not significant in low-HDI countries. In contrast, the interaction between trust in

government and HDI was not statistically significant ($B = -0.19$, 95% CI: $[-0.62, 0.24]$, $p = .388$; see Supplementary Table S6).

The interaction between workplace trust and SI was statistically significant ($B = -0.009$, 95% CI: $[-0.015, -0.004]$, $p < .001$), suggesting that the negative association between workplace trust and depressive symptoms became stronger as stringency increased. Probing this interaction revealed that among individuals in countries with relatively low stringency ($-1$ SD, SI = 46.5), higher workplace trust was associated with a modest reduction in depressive symptoms ($B = -0.49$, SE = 0.04, 95% CI: $[-0.57, -0.42]$). This negative association was stronger at average stringency levels (SI = 56.7; $B = -0.59$, SE = 0.03, 95% CI: $[-0.65, -0.53]$) and strongest in countries with high stringency ($+1$ SD, SI = 66.9; $B = -0.69$, SE = 0.04, 95% CI: $[-0.77, -0.61]$). In contrast, the interaction between trust in government and SI was not statistically significant ($B = 0.003$, 95% CI: $[-0.002, 0.009]$, $p = .249$; see Supplementary Table S6).

No multiple testing correction was applied, as interaction terms were limited and theory-driven.

## Discussion

This large, multi-country, cross-sectional study aimed to explore the relationship between HCWs' trust in the workplace and government and self-reported depressive symptoms during the early phase of the COVID-19 pandemic. Our findings underscore a significant association between trust and depressive symptoms across multiple countries, with higher trust consistently linked to fewer depressive symptoms among HCWs. This aligns with existing literature emphasizing the protective role of trust in mitigating adverse mental health outcomes during health crises (Imai, 2020; O'Hara et al., 2020).

### Interpretation of findings

This study provides robust evidence that HCWs with higher levels of trust in both the workplace and government reported significantly lower odds of depressive symptoms during the early phase of the COVID-19 pandemic. These findings build on previous research from the HEROES project, which has shown consistently elevated rates of depression among HCWs across countries (Mediavilla et al., 2022) and highlighted structural and psychosocial factors that influence mental health during health emergencies (Moro et al., 2022; Cermakova et al., 2023; Asaoka et al., 2024). Our results extend this body of evidence by emphasizing governmental and workplace trust as a cross-national protective factor for HCW mental health.

The observed association between trust and depressive symptoms aligns with the broader literature on pandemic response, including studies showing that trust in government is linked to reduced psychological distress and more adaptive behaviors during crises (Bollyky et al., 2022; Brun et al., 2022). In healthcare settings specifically, recent work in low-resource contexts has shown that institutional trust enhances HCWs' sense of preparedness, safety and psychological well-being (El Sharif et al., 2022). The present study confirms these patterns across a large and diverse international sample, suggesting that trust may serve as a form of perceived organizational and social support, buffering against the adverse mental health effects of working under prolonged uncertainty.

Importantly, our findings also highlight cross-country variability in the strength of this association. While the protective effects of trust were directionally consistent across settings, random slope models revealed significant heterogeneity across countries, likely reflecting differences in pandemic severity, national leadership, workplace culture and institutional responsiveness.

Regarding moderating factors, our findings reveal a more context-dependent role of trust in relation to country-level indicators, governmental stringency and HDI. Contrary to some prior research suggesting that more stringent public health measures may heighten psychological distress among those who trust their government (O'Hara et al., 2020), we found that workplace trust became more strongly protective against depressive symptoms as governmental stringency increased. This suggests that in environments with more intensive restrictions, often associated with heightened institutional demands and reduced autonomy, trust in one's immediate organization may serve as a critical psychological buffer.

Similarly, the protective association between workplace trust and depressive symptoms was strongest in higher-HDI countries, indicating that the benefits of institutional trust may be more pronounced in better-resourced systems, where expectations of organizational reliability and support are greater. Contrary to initial expectations, no such moderation was observed for trust in government, which may reflect differing psychological functions of these two forms of trust. While government trust may shape broader attitudes and compliance behaviors, workplace trust likely plays a more immediate and emotionally salient role for HCWs navigating day-to-day stressors. These patterns emphasize the importance of fostering trust at the institutional level, particularly in high-demand or highly regulated settings, and point to the need for further research into how structural conditions shape the mental health impact of trust during global health emergencies (Lee et al., 2021).

### Strengths and limitations

Our study represents a significant contribution to understanding the relationship between trust in the workplace and government and depressive symptoms among HCWs during the COVID-19 pandemic. With a large, multinational sample comprising 32,410 HCWs across 22 countries, our study offers a comprehensive analysis of how trust influences mental health outcomes in diverse healthcare settings. By including HCWs from various professional backgrounds and geographic regions, we enhance the generalizability of our findings and provide insights into cross-cultural variations in trust and its impact on depressive symptoms during crises. Furthermore, we used rigorous statistical methods to analyze the association between trust variables and depressive symptoms, adjusting for individual and country-level factors. Overall, our study contributes valuable evidence supporting the role of trust as a protective factor against depressive symptoms among HCWs, informing future interventions and policies aimed at bolstering healthcare workforce resilience during pandemics and other public health emergencies.

However, several limitations should be acknowledged in our study. First, while we captured a snapshot of the association between the SI and depressive symptoms among HCWs, our design did not allow for capturing the dynamic changes in governmental stringency policies, COVID-19 metrics and HCWs' mental health over time. The complex and bidirectional relationship between governmental trust and mental health, as highlighted in prior research (Jakovljevic et al., 2020), suggests the possibility of reverse causation and reverse buffering, where HCWs experiencing depressive symptoms may be more inclined to distrust governmental responses.

Second, variation in recruitment timing across countries, combined with the use of nonprobabilistic or snowball sampling in most

sites, may have introduced selection and sampling biases. These limitations are compounded by the potential underrepresentation of HCWs most burdened by the pandemic (e.g., those with high workloads or direct COVID-19 care responsibilities). Furthermore, due to the absence of a formal nonresponse analysis and the lack of a weighting scheme, the generalizability of findings to national HCW populations should be interpreted with caution.

Third, while missing data were addressed using multiple imputation, the assumption of data MAR was made based on observed patterns and auxiliary variable inclusion. However, Little's MCAR test indicated that the data were not missing completely at random, and the use of imputation may not fully eliminate potential bias.

Finally, we relied on single-item measures to assess trust in the workplace and government. While single-item instruments offer simplicity and reduce respondent burden, they may fail to capture the multifaceted nature of trust, making them less robust than multi-item scales that provide a more nuanced evaluation of trust perceptions (Grimmelikhuijsen and Knies, 2015; Castro et al., 2023). In political and organizational research, single-item trust measures have been widely used but also criticized for their limited depth and potential measurement error (Marien, 2011). Moreover, although depressive symptoms were assessed using the validated PHQ-9 instrument, we did not conduct formal measurement invariance analyses across countries for either the trust items or the PHQ-9 scale. This is a notable limitation in cross-national survey research, where cultural and linguistic differences may affect how respondents interpret and respond to items. Future research should consider using multi-item, cross-culturally validated trust instruments and formally test for measurement invariance to strengthen the robustness and comparability of international findings.

### Lessons learned

This study underscores a key lesson: institutional and governmental trust serve as critical psychological buffers for HCWs across diverse settings. However, it is not generalized or blind trust that protects mental health, but rather trust earned through consistent, transparent and supportive institutional behaviors. Prior studies point to inclusive decision-making, visible leadership and proactive communication as key components that foster such trust (De Kock JH et al., 2022; Robins-Browne et al., 2022).

While some trust-building initiatives may require substantial resources, our findings are equally relevant to low-HDI settings, where feasible, low-cost strategies can have a meaningful impact. Examples include participatory safety climate committees, leadership walkarounds, transparent reporting of personal protective equipment availability, inclusive dialogue and fair allocation of resources. These practices have been shown to enhance institutional trust even in crisis contexts and can be implemented without major structural investments (De Kock JH et al., 2022; Robins-Browne et al., 2022). Embedding such approaches into routine organizational culture may support HCW resilience not only during pandemics but also in the face of persistent healthcare system challenges.

Future research should prioritize longitudinal designs, standardized trust measures and probabilistic sampling to clarify the causal relationships between trust, institutional responses and HCW mental health outcomes. Multifaceted strategies that address both individual-level stressors and systemic trust-building mechanisms will be critical to strengthening healthcare workforce resilience in the long term.

### Conclusion

This study underscores the critical importance of trust, particularly in the workplace, as a protective factor against depressive symptoms among healthcare workers during the COVID-19 pandemic. While both workplace and government trust were associated with less depressive symptoms, only workplace trust demonstrated stronger protective effects in more developed countries and under stricter government responses, highlighting its unique role in high-pressure settings. Notably, the strength of these associations varied across countries, reflecting the influence of national and institutional context. Cultivating workplace trust should be a central component of mental health support strategies for HCWs, especially during times of prolonged crisis and uncertainty.

**Open peer review.** To view the open peer review materials for this article, please visit http://doi.org/10.1017/gmh.2025.10067.

**Supplementary material.** The supplementary material for this article can be found at http://doi.org/10.1017/gmh.2025.10067.

**Data availability statement.** The code used for the analyses in this article is available on request. For the use of data, proposals can be sent to e.m.a.vander.ven@vu.nl.

**Acknowledgements.** The authors would like to thank all the healthcare workers who filled in our survey; without them, this work would not be possible.

**Author contribution.** DB, DC, and EV were responsible for the study conceptualization, supervision and writing the original manuscript draft. MF, RJ, AAB, AI, LG, AB, DB, MTC, AT, JR, ARH, JN, JST, AM, NS, JS, JL, VPP, MPA, MFM, MGC, NK, DN, EK, GK, DS, SDA, JCH, OG, OA, AD, ER, LA, SA, MBO, GM-A, EFJ, RMT, EV, UO, DK and AMR were involved in collecting and coordinating data collection at different healthcare centers. ES, RA and FM initiated the international HEROES study and set up the study framework. All authors contributed to the interpretation of results, reviewed preliminary versions of the manuscript and approved the final version.

**Financial support.** The Czech COVID-19 HEROES project was funded by the Ministry of Health of the Czech Republic (NU22J-09-00064, PI: Dominika Seblova). GM-A is partially supported by the Fundacion Martin Escudero. The HEROES project in Colombia was funded by the Minciencias (CT 860-2020) and by the UK Research and Innovation (UKRI; ES/V013157/1). The HEROES project in Italy was funded by the Italian Ministry of Education, Universities and Research (Bando FISR 2020IP_05308), and Fondazione di Sardegna (Bando 2020). The HEROES project in Japan was funded by the Japanese Ministry of Health, Labour and Welfare (19IA2014).

**Funding statement.** Open access funding provided by Vrije Universiteit Amsterdam

**Competing interests.** The authors declare none.

**Ethics statement.** The study was approved by accredited Institutional Review Boards (IRBs) in each participating country, including the IRB of the Faculty of Medicine, University of Chile; Columbia University Medical Center IRB; Hospital La Paz Ethics Committee; Research Ethics Committee of the Graduate School of Medicine and Faculty of Medicine at the University of Tokyo; National Committee of Ethics in Research (CONEP/CNS/MS); Ethics Committee of the Azienda Mista Ospedaliero-Universitaria of Cagliari; University College Hospital Ethics Review Committee, University of Ibadan; Instituto Jalisciense de Salud.
Mental (SALME); Comité de Ética de Investigación en Seres Humanos, de la Facultad de Medicina, Universidad de Chile; IRB University of Applied Sciences Emden / Leer; Comité de Bioética, de la Facultad de Medicina, Universidad Mayor de San Simón; Ponce Health Sciences University; Ethics Committee of the Czech Ministry of Health, Ethics Committee at the Second Faculty of Medicine; Institutional Ethics Committee at the Institute of Molecular Biology of National Academy of Sciences of Armenia; King Abdullah International

Medical Research Center (KAIMRC) IRB committee; Universidad de San Carlos de Guatemala Comité de Bioética en Investigación en Salud; Bioethics Committee of the National School of Public Health, University of Antioquia and Pan-American Health Organization Ethics Review Committee. In the Netherlands, ethical approval was waived by the Medical Ethical boards of the Maastricht University Medical Center and the Amsterdam Medical Center, considering that the participants were not regarded as patients or identifiable persons sharing sensitive information, following the Medical Research Involving Human Subjects Act (WMO). The procedures used in this study adhere to the tenets of the Declaration of Helsinki.

**Informed consent.** Informed consent was obtained from all individual participants included in the study.

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

## Appendix

**Table A1.** Individual-level characteristics stratified by trust in the workplace and government

| | All (N = 32,410) | Trust in the workplace | | Statistical tests | Trust in the government | | Statistical tests |
|---|---|---|---|---|---|---|---|
| | | Low (n = 8,337) | High (n = 20,128) | $\chi/t$, p-value | Low (n = 14,775) | High (n = 13,659) | $\chi/t$, p-value |
| Age, mean (SD) | 39.8 (11.2) | 39.7 (10.6) | 40.1 (11.4) | $t(19,653) = -3.64$, $p = .000$ | 38.8 (10.6) | 41.3 (11.6) | $t(5,982) = -13.91$, $p < .001$ |
| Missing, n (%)[1] | 4,847 (15) | 680 (20) | 2,706 (80) | | 1,631 (48) | 1,744 (52) | |
| Gender, n (%) | | | | $\chi^2(2) = 25.15, p < .001$ | | | $\chi^2(2) = 12.91, p = .002$ |
| Woman | 23,167 (72) | 6,294 (30) | 14,737 (70) | | 11,046 (53) | 9,958 (47) | |
| Man | 8,140 (25) | 2,019 (27) | 5,360 (73) | | 3,699 (50) | 3,676 (50) | |
| Other | 58 (0) | 17 (41) | 24 (59) | | 22 (54) | 19 (46) | |
| Missing | 1,045 (3) | 7 (50) | 7 (50) | | 8 (57) | 6 (43) | |
| Education level, n (%) | | | | $\chi^2(5) = 88.31, p < .001$ | | | $\chi^2(5) = 62.19, p < .001$ |
| Primary school | 265 (1) | 65 (26) | 151 (74) | | 114 (51) | 101 (49) | |
| Secondary school | 1,958 (6) | 383 (22) | 1,377 (78) | | 803 (46) | 957 (54) | |
| Technical-professional | 4,461 (14) | 1,071 (26) | 2,971 (73) | | 2,091 (52) | 1,939 (48) | |
| Undergraduate | 11,183 (34) | 3,156 (31) | 7,007 (69) | | 5,493 (54) | 4,655 (46) | |
| Postgraduate | 13,397 (41) | 3,658 (30) | 8,600 (70) | | 6,261 (51) | 5,994 (49) | |
| Missing | 1,146 (3) | 4 (15) | 22 (85) | | 13 (50) | 13 (50) | |
| Current job, n (%) | | | | $\chi^2(5) = 327.83$, $p < .001$ | | | $\chi^2(5) = 283.63$, $p < .001$ |
| Physician | 8,869 (27) | 2,837 (34) | 5,613 (66) | | 4,710 (56) | 3,738 (44) | |
| Nurse | 6,568 (20) | 1,980 (32) | 4,193 (68) | | 3,230 (52) | 2,936 (48) | |
| Health technicians | 3,130 (10) | 697 (24) | 2,242 (76) | | 1,516 (52) | 1,417 (48) | |
| Ancillary workers | 3,989 (12) | 872 (23) | 2,927 (77) | | 1,854 (49) | 1,941 (51) | |
| Other HCWs | 7,407 (22) | 1,943 (28) | 5,109 (72) | | 3,450 (49) | 3,590 (51) | |
| Missing | 2,447 (8) | 8 (15) | 44 (85) | | 15 (29) | 37 (71) | |
| Contact with COVID−19 patients, n (%) | | | | $\chi^2(1) = 7.67, p = .006$ | | | $\chi^2(1) = 342.33$, $p < .001$ |
| Yes | 14,434 (45) | 4,055 (29) | 10,029 (71) | | 7,945 (56) | 6,120 (44) | |
| No | 9,711(30) | 2,530 (29) | 6,906 (71) | | 4,077 (43) | 5,350 (57) | |
| Missing | 8,265 (25) | 1,752 (35) | 3,193 (65) | | 2,753 (56) | 2,189 (44) | |
| Depressive symptoms [PHQ−9 ≥ 10], n (%) | | | | $\chi^2(1) = 204.67$, $p < .001$ | | | $\chi^2(1) = 216.76$, $p < .001$ |
| No | 19,496 (60) | 5,239 (27) | 14,247 (73) | | 9,445 (48) | 10,034 (52) | |
| Yes | 5,421 (17) | 2,009 (37) | 3,411 (63) | | 3,440 (63) | 1,980 (37) | |
| Missing | 7,493 (23) | 1,090 (13) | 2,470 (12) | | 1890 (13) | 1,646 (12) | |

[1]All percentages are valid.
*Note:* HCW, healthcare worker; PHQ-9, Patient Health Questionnaire − 9 items.
Exact *p*-values are reported unless <.001, in which case "<.001" is shown for readability.

**Table A2.** Odds ratios and 95% confidence intervals (OR [95% CI]) for depressive symptoms among low and high trust in the workplace and government per country stratified by HDI (*N* = 32,410)

| | Trust in the workplace[1] | | Trust in the government[1] | |
|---|---|---|---|---|
| | Unadj. OR (95% CI) | Adj. OR[2] (95% CI) | Unadj. OR (95% CI) | Adj. OR[2] (95% CI) |
| All | 1.43** | 1.38** | 1.40** | 1.38** |
| | (1.36–1.50) | (1.32–1.46) | (1.34–1.47) | (1.31–1.44) |
| Countries | | | | |
| Puerto Rico[3] | 2.00* | 2.00* | .93 | . 87 |
| | (1.10–3.63) | (1.09–3.68) | (.54–1.60) | (.50–1.52) |
| Low HDI | | | | |
| Nigeria (.547) | .69 | .79 | 1.64* | 1.59* |
| | (.46–1.03) | (.53–1.20) | (1.10–2.44) | (1.05–2.39) |
| Medium HDI | | | | |
| Guatemala (.645) | 1.32* | 1.21 | 1.23 | 1.07 |
| | (1.08–1.62) | (.98–1.49) | (.99–1.53) | (.86–1.35) |
| Bolivia (.697) | 1.035 | 1.02 | .88 | .87 |
| | (.53–2.02) | (.51–2.02) | (.47–1.65) | (.46–1.65) |
| Venezuela (.699) | 1.06 | 1.06 | .92 | .91 |
| | (.83–1.38) | (.82–1.36) | (.72–1.19) | (.70–1.17) |
| High HDI | | | | |
| Tunisia (.733) | .93 | .95 | .83 | .78 |
| | (.678–1.28) | (.69–1.32) | (.58–1.19) | (.54–1.13) |
| Lebanon (.750) | 3.21** | 3.27** | 1.14 | 1.10 |
| | (2.15–4.79) | (2.19–4.90) | (.83–1.56) | (.80–1.51) |
| Armenia[4] (.761) | 1.82* | 2.02** | 1.98** | 1.94** |
| | (1.24–2.68) | (1.36–3.01) | (1.39–2.81) | (1.35–2.78) |
| Mexico (.763) | .96 | .95 | 1.07 | 1.04 |
| | (.83–1.11) | (.82–1.10) | (.91–1.29) | (.89–1.23) |
| Colombia (.764) | 1.60* | 1.59* | 1.51* | 1.42* |
| | (1.07–2.41) | (1.05–2.40) | (1.08–2.10) | (1.01–1.99) |
| Peru (.769) | 1.20* | 1.19* | 1.31** | 1.33** |
| | (1.04–1.38) | (1.04–1.37) | (1.15–1.50) | (1.16–1.52) |
| Brazil (.770) | 2.13** | 2.25** | 1.00 | 1.02 |
| | (1.78–2.55) | (1.87–2.70) | (.87–1.15) | (.88–1.18) |
| Very high HDI | | | | |
| Argentina (.851) | 1.30 | 1.29 | 1.57** | 1.49* |
| | (.99–1.70) | (.98–1.70) | (1.22–2.02) | (1.16–1.93) |
| Chile (.856) | 2.15** | 2.02** | 1.52** | 1.37** |
| | (1.74–2.65) | (1.63–2.51) | (1.29–1.79) | (1.16–1.63) |
| Saudi Arabia (.875) | 2.55* | 2.53* | 4.75** | 5.14** |
| | (1.37–4.73) | (1.34–4.77) | (2.06–10.96) | (2.18–12.09) |
| Czech Republic (.898) | 2.04** | 1.88** | 1.85** | 1.74** |
| | (1.56–2.66) | (1.43–2.46) | (1.49–2.29) | (1.39–2.17) |
| Italy (.899) | 1.61** | 1.60** | 1.71** | 1.67** |
| | (1.44–1.81) | (1.43–1.80) | (1.53–1.91) | (1.49–1.87) |

(*Continued*)

**Table A2.** (*Continued*)

| | Trust in the workplace[1] | | Trust in the government[1] | |
|---|---|---|---|---|
| | Unadj. OR (95% CI) | Adj. OR[2] (95% CI) | Unadj. OR (95% CI) | Adj. OR[2] (95% CI) |
| Spain (.901) | 1.51** | 1.47** | 1.19* | 1.17 |
| | (1.25–1.82) | (1.21–1.78) | (1.01–1.39) | (.99–1.37) |
| Japan (.922) | 2.14* | 1.79* | 2.24* | 1.95* |
| | (1.24–3.68) | (1.03–3.12) | (1.34–3.72) | (1.16–3.30) |
| Belgium (.939) | 2.17* | 2.21* | 1.12 | 1.14 |
| | (1.01–4.67) | (1.02–4.80) | (.61–2.07) | (.62–2.13) |
| Netherlands (.945) | 2.58** | 2.62** | 2.61** | 2.61** |
| | (1.66–4.01) | (1.68–4.10) | (1.71–3.97) | (1.70–4.00) |
| Germany (.955) | 3.05* | 2.87* | 6.98** | 6.65** |
| | (1.31–7.14) | (1.19–6.91) | (3.23–15.05) | (3.02–14.65) |

*Note:* Unadj. OR, unadjusted OR; adj. OR, adjusted OR; HDI, Human Development Index – A composite index measuring average achievement in three basic dimensions of human development: a long and healthy life, knowledge and a decent standard of living. Very high human development (≥0.800), high human development (0.700–0.799), medium human development (0.550–0.699) and low human development (<0.550).
Exact *p*-values are reported unless <.001, in which case "<.001" is shown for readability. Asterisks indicate statistical significance: *p* < .05 (*), *p* < .001 (**).
[1]Reference category is 1 = high trust.
[2]Adjusted OR for age, gender, current job and exposure.
[3]Adjusted OR was calculated without the HDI value since Puerto Rico is an unincorporated US territory.
[4]Adjusted OR was calculated without the SI value.