## [Reviewer Report]

This study is based on the first wave of the international HEROES survey, an online cross-sectional assessment conducted between March 2020 and February 2021 among 32,410 healthcare workers from 22 countries covering the full range of national income levels. The PHQ-9 is used to identify clinically relevant depressive symptoms. The authors tested whether two single-item indicators of trust - one directed at the workplace, the other at the national government - predicted lower odds of depression after adjusting for sociodemographic, occupational, and pandemic-related covariates. Multilevel logistic models show that high trust in the workplace and high trust in the government are each associated with about a 28% reduction in the odds of depressive symptoms. Although cross-level interactions with the Human Development Index and the Oxford Stringency Index are examined, they are small and do not survive correction for multiple testing. The authors conclude that institutional and organizational trust are modifiable protective factors for mental health during large-scale crises. In this sense, it’s a shame that the authors did not pre-register their analyses.

The manuscript addresses a policy-relevant question with an exceptionally large and geographically diverse sample. The finding that trust functions may be a robust correlate of mental health independent of known risk factors can be used for interventions beyond individual-level resilience training. The analytic strategy is appropriate-although I have some issues with the dichotomization of the items-the writing is clear, and the discussion acknowledges most of the important caveats. While the cross-sectional design limits causal inference, the sheer breadth of the data set and the multilevel perspective significantly advance the literature. Therefore, I am inclined to recommend publication after revision, provided the authors address the concerns outlined below.

Major remarks

The manuscript would benefit from a clearer description of the temporal alignment between the survey window and the evolution of national pandemic indicators, as trust can fluctuate rapidly with policy changes and case surges. The decision to dichotomize the single-item trust questions is statistically unsound (see Irwin & Mclelland, 2003), and analyses should always be conducted on continuous variables when available, unless there is a strong theoretical rationale for dichotomizing them. At the very least, sensitivity analyses using the original four-point scale, or at least an alternative cut point, would reassure readers that loss of information has not biased effect sizes and increased type I errors.

Moreover, the trust constructs are measured with ad hoc single items that have not been validated across cultures. The paper would benefit from reporting basic psychometric information-item distributions, country-level endorsement patterns, and tests of measurement invariance where possible-or at least acknowledging the limits such brevity imposes on construct validity. If comparable multi-item trust scales exist in the HEROES questionnaire, presenting a brief robustness check with these alternatives would greatly increase confidence in the conclusions.

Since missing data were handled by multiple imputation, it would be useful to report the proportion of missing data per variable and to compare imputed versus complete case results in an online appendix. It would also be an improvement to add whether the data were missing at random or not (i.e., MAR, MCAR) to check whether imputation and what type of imputation is appropriate. Given the nonprobabilistic sampling and the over-representation of certain occupational groups in some countries, a brief analysis of non-response or a weighting scheme would strengthen claims of generalizability.

The main problem with the study is its observational design. The potential for reverse causality - depressed mood reduces perceived trust - should be discussed more fully, ideally with reference to existing longitudinal evidence. Although the authors mention that the cross-sectional design prevents causal inference, the discussion could more fully address potential bidirectional or reciprocal relationships. In addition, the inclusion of the concept of “reverse buffering” - whereby depressed individuals perceive institutions less favorably - would enrich the theoretical framework.

Currently, country-level variance components are only reported for the final models; presenting intraclass correlation coefficients for empty models would help readers assess how much of the outcome heterogeneity is at the national level before adding predictors.

Relatedly, the PHQ-9 has been translated into many languages, but does its optimal cut-off vary across cultural settings? A supplementary table documenting which validated versions have been used in each country and confirming that a cut-off of ten is appropriate everywhere would dispel doubts about differential misclassification of depression.

The multilevel models include random intercepts by country, but do not clarify whether random slopes for trust were examined. Given the theoretical expectation that the effect of trust might vary across institutional contexts, testing for random slopes (and reporting model fit comparisons) would indicate whether the protective effect is truly homogeneous. If convergence issues preclude such models, it would be helpful to state this explicitly. However, given the sample size, I think the models would converge.

With respect to control variables, the rationale for including some but not others is not always clear. For example, educational attainment, occupational income level, and prior mental health diagnoses could plausibly confound the relationship between trust and depression; if these were not available, their omission should be discussed. This is related to my earlier comment about pre-registering analyses and design before conducting the study.

The examination of macro-level moderators relies on the Human Development Index and the Oxford Stringency Index, both of which are composites. Authors should justify why the HDI, rather than gross national income or health expenditure per capita, best captures structural capacity relevant to trust. Similarly, the use of daily, weekly, or monthly values of the Stringency Index during the survey period needs to be specified, since fluctuations in restrictions could affect both trust and sentiment.

Statistical reporting can be streamlined by reporting confidence intervals for all fixed effects in the main text rather than in the Supplement, giving readers an immediate sense of precision. The current use of a single p-value threshold (<.001) for many predictors risks giving a false impression of uniform certainty; exact p-values (or at least three decimals) would be preferable. For the multiple testing issue raised by the interaction models, please indicate whether a correction (e.g., Bonferroni, Benjamini-Hochberg) was applied.

The section on practical implications is compelling, but could be more nuanced for low-HDI settings where organizational resources for trust-building interventions may be scarce; suggesting context-appropriate, low-cost strategies would broaden the article’s impact.

Finally, the discussion positions trust as a “modifiable” factor, but practical pathways remain vague. The authors could enrich their applied message by citing successful organizational interventions - such as participatory safety climate committees, leadership walk arounds, or transparent PPE allocation dashboards - that have been shown to increase trust in similar crises.

Irwin, J. R., & McClelland, G. H. (2003). Negative Consequences of Dichotomizing Continuous Predictor Variables. Journal of Marketing Research, 40, 366-371.

---

## [Reviewer Report]

This study examines the relationship between healthcare workers’ (HCWs) trust in their workplace and the government and depressive symptoms during the COVID-19 pandemic. The following issues should be addressed:

The introduction section is overly fragmented and should be restructured for better coherence and logical progression.

Table 1 should incorporate statistical tests to strengthen the descriptive analysis.

A multilevel regression analysis is needed to assess potential country-level effects.

The data analysis is superficial, particularly regarding interaction effects, which tables are not shown.

The discussion section is underdeveloped and requires further elaboration.

---

## [Reviewer Report]

This article is very well written and researched, I recommend it for publication with some minor revisions.

The association between health worker trust in governments and workplaces and reduced likelihood of depressive symptoms is well articulated. To further strengthen the article, the authors might consider expanding on the characteristics that foster trustworthiness in these institutions. For example, trust in the workplace is likely built through transparent leadership communication, inclusive decision-making processes, and visible efforts to support staff well-being. These elements contribute to a more positive work environment, which improves mental health outcomes. This distinction reinforces that it is not generalized or blind trust that is protective, but rather trust earned through care of staff well-being, positive working environments, and supportive policies.

When the Stringency Index is introduced on page 7, it would be helpful to add a brief description of it, since it is lesser known than the HDI.

Thank you for the opportunity to review.

---

## [Reviewer Report]

Overall, I am satisfied with the reviewer’s response, but I have a few questions that require clarification.

In my previous comments, I mentioned that the authors should report item distributions and endorsement profiles by country. If available, they should also include a test with multi-item scales to check for possible invariance or, at the very least, provide a more robust list of limitations. Although the authors included sections reminding readers of the constraints of a multinational survey and a paragraph on limitations, which is an improvement, there are still no tables showing distribution by country, no invariance test, and no multi-item alternatives. If the authors did not collect this data, that is not a major issue, but they should mention it.

Additionally, the authors should standardize the p-values in the tables. Currently, some lines are in the format “* p <.05, ** p <.001,” while others have exact p-values. Reporting the exact p-values is preferable. This would facilitate future meta-analyses.